# Cell type-specific mapping of ion distribution in *Arabidopsis thaliana* roots

Ricardo F. H. Giehl [1] ✉, Paulina Flis [2], Jörg Fuchs [1], Yiqun Gao [2], David E. Salt [2] & Nicolaus von Wirén [1] ✉

Cell type-specific mapping of element distribution is critical to fully understand how roots partition nutrients and toxic elements with aboveground parts. In this study, we developed a method that combines fluorescence-activated cell sorting (FACS) with inductively coupled plasma mass spectrometry (ICP-MS) to assess the ionome of different cell populations within *Arabidopsis thaliana* roots. The method reveals that most elements exhibit a radial concentration gradient increasing from the rhizodermis to inner cell layers, and detected previously unknown ionomic changes resulting from perturbed xylem loading processes. With this approach, we also identify a strong accumulation of manganese in trichoblasts of iron-deficient roots. We demonstrate that confining manganese sequestration in trichoblasts but not in endodermal cells efficiently retains manganese in roots, therefore preventing toxicity in shoots. These results indicate the existence of cell type-specific constraints for efficient metal sequestration in roots. Thus, our approach opens an avenue to investigate element compartmentation and transport pathways in plants.

Besides supplying above-ground tissues with water and mineral elements, plant roots also play a critical role in selecting, enriching, and retaining mineral elements to adjust the type and quantity of elements demanded by the shoot. To execute such element-specific functions, roots express a range of ion transporters that mediate the uptake, efflux and intracellular compartmentalization of different mineral elements. Most ion transporters show characteristic tissue and cell type-specific localization patterns, which strongly depend on root developmental stage and changes in internal signaling and external cues[1–4]. Such discrete localization patterns are thought to distinctly distribute different elements along the longitudinal and radial axes of a root, and determine the most prominent transport pathways used by each element. However, assessing and understanding the link between the localization of ion transporters and transport pathways is challenging due to a lack of specificity of many transporters for a single element, and the difficulty in quantifying element concentrations at the cellular level.

Over the last decades, a series of bioimaging techniques have been developed to investigate element distribution in plant tissues, which include x-ray fluorescence microscopy (XRF), particle-induced X-ray emission (microPIXE) and nanoscale secondary ion mass spectroscopy (NanoSIMS)[5–7]. However, these methods are typically only semi-quantitative, entail complex sample preparation, or require highly specialized equipment not available to most laboratories. The distribution of some elements can also be visualized with the help of element-sensitive dyes[8] or fluorophores[9], or with genetically encoded fluorescent biosensors (reviewed in refs. [10], [11]). However, the specificity of each chemical or molecular sensor does not allow multi-element analysis of the same tissue, and the sensitivity varies largely from sensor to sensor. The best alternative for simultaneous, absolute quantification of most elements of the periodic table with very high sensitivity is inductively coupled plasma mass spectrometry (ICP-MS). This technique has been successfully employed to carry out high-throughput multi-element profiling of mutants and genetically diverse natural accessions of various plant species, leading to the discovery of several genes regulating a plant's ionome (reviewed in ref. [12]). However, despite its high sensitivity, ICP-MS analysis is still largely confined to whole-tissue samples. This limitation is due to the necessity to

[1]Leibniz-Institute of Plant Genetics and Crop Plant Research (IPK) OT Gatersleben, 06466 Seeland, Germany. [2]Future Food Beacon of Excellence & School of Biosciences, University of Nottingham, Nottingham LE12 5RD, UK. ✉e-mail: giehl@ipk-gatersleben.de; vonwiren@ipk-gatersleben.de

introduce the biological sample into the plasma of the ICP-MS device, most commonly achieved by fully digesting the tissue in an acid solution prior to sample injection. To circumvent this constraint, a laser can be used to dissect regions of interest from fixed tissue[5,7,13–16]. In this way, the laser-induced aerosol is directly transferred into the ICP-MS allowing spatially resolved mapping of multiple elements in selected cells or tissues. Although laser ablation coupled with ICP-MS (LA-ICP-MS) was originally restricted to the analysis of artificially dehydrated samples or naturally dry tissue[15,17,18], improved sample preparation has recently enabled the use of this technique to map elemental distribution in fresh roots of barley and *Arabidopsis thaliana* plants[14,19]. However, element levels in the symplast and apoplast cannot be clearly distinguished with this technique.

One alternative to laser-assisted tissue microdissection may be the use of fluorescence-activated cell sorting (FACS) prior to ICP-MS analysis to separate distinct cell types marked with the expression of a fluorescent protein (FP). Previous studies have already successfully used sorted protoplasts to assess the transcriptome[20,21], small RNA profiles[22], proteome[23], and DNA methylome[24] of *A. thaliana* roots with high spatial resolution. FACS-based protoplast sorting has also been combined with high-resolution mass spectrometry for the analysis of metabolites[25] and plant hormones[26,27]. Together, these studies revealed the existence of cell type-specific gene expression and protein accumulation patterns that ultimately result in spatial compartmentation of many biological processes and in discrete distributions of several metabolites and plant hormones. Interestingly, transcriptomics of sorted FP-expressing protoplasts representing all radial zones of an *A. thaliana* root exposed to iron (Fe) deficiency revealed that groups of genes involved with Fe uptake, storage and signaling responded to deficiency in a cell type-specific manner[3]. Therefore, the development of FACS-assisted ICP-MS analysis could provide an exciting possibility to link the elemental composition of a specific cell population with its transcriptome and proteome.

In this study, we established a method for multi-element analysis of sorted protoplasts originating from distinct cell types of *A. thaliana* roots. We used the new FACS-ICP-MS method to reveal significant cell type-specific element distribution and the existence of a steep concentration gradient between outer and inner cell layers in roots. Furthermore, our method uncovered previously unknown changes in element distribution caused by disturbed xylem loading processes, and identified prominent manganese (Mn) accumulation in root hair cells when plants are exposed to limited Fe availability. Our results also demonstrate a high degree of cell type-specific dependence of vacuolar loading for efficient Mn sequestration in roots.

## Results

### Methodology to obtain contaminant-free protoplasts for ICP-MS analysis

Compared to other approaches relying on protoplast isolation for downstream analysis, ionomics of cells isolated using conventional methods is prone to contamination by various elements in buffers, labware and reagents used during cell wall digestion (Supplementary Fig. 1a). To decrease the interference of these contaminants, we first evaluated if protoplast yield and stability can be maintained if cell wall-hydrolyzing enzymes were resuspended in a solution containing only pH-buffered mannitol. To avoid significant changes in elemental composition due to ion influx and efflux occurring during isolation, we restricted protoplast isolation to 90 min. Within this time period ~250,000 protoplasts could be isolated from roots of ~3000 5-day-old *A. thaliana* plants. No significant losses of size, yield and protoplast stability and integrity were observed when the standard salt solution was replaced by the salt-free solution (Fig. 1a and Supplementary Fig. 1b–d). We also detected significant levels of several elements in the lyophilized enzyme powder (Supplementary Table 1), which should be removed prior to ICP-MS analysis. By testing subsequent washes of

the pelleted cells with pure 5% mannitol we determined that two washes were sufficient to remove almost all contamination left from the protoplast isolation solution (Supplementary Fig. 2).

To assess if the developed procedure was able to reliably detect inherent ionomic changes in isolated root cells, we compared culture-dependent elemental concentrations in whole roots and in isolated protoplasts. Therefore, *A. thaliana* seedlings (Col-0) were cultivated on solid media with full nutrient supply (control), or depleted of potassium (K) or phosphorus (P) as described previously[28]. In all cases, growth media were supplemented with subtoxic levels of rubidium (Rb), strontium (Sr) and selenium (Se). Under these conditions, K or P levels were significantly decreased in roots of 5-day-old seedlings grown in K- or P-depleted medium, respectively (Supplementary Fig. 3a). These decreases were more pronounced at the whole-root level than in protoplasts. Importantly, we detected an expected increase in the levels of Rb, a chemical analog of K, specifically in K-deficient roots as well as protoplasts (Fig. 1b, c). Furthermore, we observed a significant increase of calcium (Ca) and magnesium (Mg) levels in whole roots but not in protoplasts of K-deficient plants (Supplementary Fig. 3a), confirming that binding of $Ca^{2+}$ and $Mg^{2+}$ ions to the cell wall is enhanced in the absence of $K^+$[29]. When element accumulation in protoplasts was referred to whole-root levels, it was possible to obtain a proxy for element distribution between the symplast and the apoplast. In line with the primary function of boron (B) in cross-linking rhamnogalacturonan II molecules in the cell wall[30], this micronutrient accumulated at comparatively low levels in protoplasts and had a low symplast accumulation ratio (Supplementary Fig. 3a, b). Relatively low enrichment in protoplasts was also detected for Mn, Fe and Sr, indicating that in whole-roots the bulk amount of these elements was located in the cell wall. In contrast, Mg and P were more enriched in protoplasts, except when plants were grown in K-deficient media (Supplementary Fig. 3b), supporting their high affinity for cell wall adsorption or precipitation[29,31]. Interestingly, zinc (Zn) levels dropped in response to P deficiency and showed higher relative partitioning to the symplast (Supplementary Fig. 3a, b).

To further test how reliably our method can detect changes in micronutrient levels, we assessed Fe, Zn and Mn in protoplasts isolated from roots of the Fe uptake-defective mutant *irt1*. The poor selectivity of IRT1 towards $Fe^{2+}$ and the up-regulation of *IRT1* expression by Fe deficiency results in the overaccumulation of various divalent metals in roots of plants grown on Fe-depleted media[32–34]. As expected, we detected less Fe in protoplasts isolated from Fe-deficient wild-type roots, while Mn and Zn levels were increased (Fig. 1d–f). However, in root protoplasts of *irt1-1* mutant plants, no significant overaccumulation of Mn and Zn was detected. When Fe was supplied to the media, only Fe and Zn concentrations were significantly lower in protoplasts isolated from roots of *irt1-1* than in protoplasts of wild-type roots (Fig. 1d–f), whereas a drop in Mn accumulation under this condition was probably prevented by the activity of the Mn uptake transporter NRAMP1[35,36]. Altogether, these results demonstrate that our procedure enables efficient isolation of contaminant-free protoplasts from roots and reliable ionomic profiling of macronutrients, micronutrients and trace elements.

### Controlling element losses during FACS

In the next step, we optimized conditions to sort protoplasts prior to ICP-MS analysis. We targeted a period of ~30 min usually required for FACS after isolation. Therefore, protoplasts were isolated and washed as described above, re-suspended in pure 5% mannitol and placed on ice. Our isolation method was able to maintain protoplast size and stability as well as the expression of a GFP reporter for more than 30 min after isolation (Supplementary Fig. 4a–c). We then assessed if elemental concentrations in protoplasts change during the period required for sorting. To do so, we enriched roots with lithium (Li), an element that is present at only ultra-trace levels (not detected by our

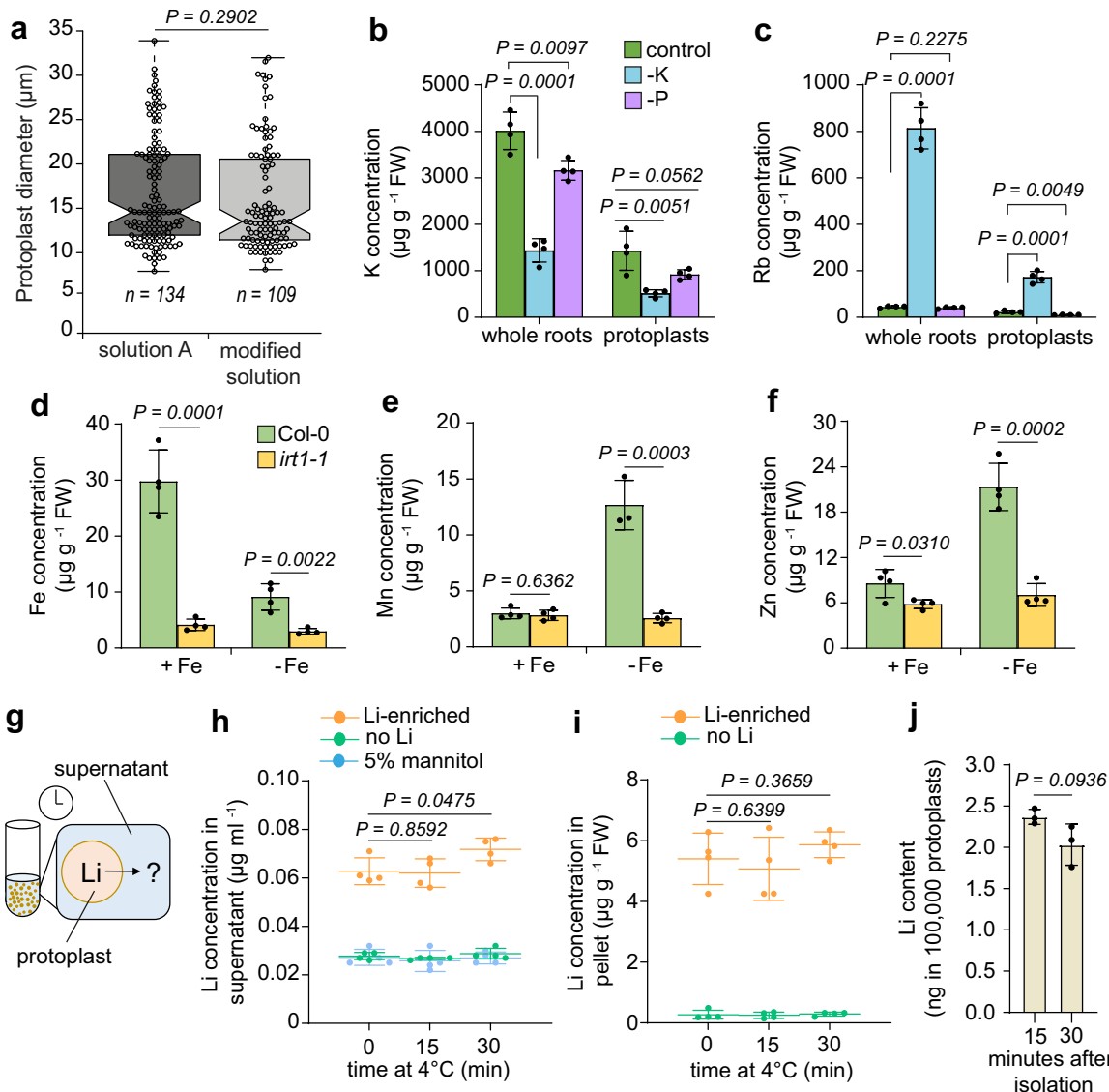

**Fig. 1 | Development of a method for reliable multi-element analysis of isolated root protoplasts. a** Comparison of standard solution A and modified protoplasting solution on the diameter of isolated root protoplasts. The boxes extend from the first to the third quartile around the median, while the ends of the whiskers indicate the maximum and minimum values within 1.5x the interquartile range from the box ends. **b, c** Concentration of K (**b**) and its chemical analog Rb (**c**) in whole roots and root protoplasts isolated from plants grown on solid media containing sufficient levels of all nutrients (control) or depleted of K or P. Bars show means ± SD ($n = 4$ biological replicates). FW, fresh weight. **d–f** Concentration of Fe (**d**), Mn (**e**), and Zn (**f**) in protoplasts isolated from roots of wild-type (Col-0) and *irt1* mutant plants grown on sufficient (+Fe) or insufficient Fe (−Fe). Bars show means ± SD ($n = 4$ biological replicates). **g–j** To estimate potential element leakage during simulated sorting, roots were enriched with Li by cultivating Col-0 plants in Li-containing agar for 5 days. Protoplasts isolated from roots of plants grown or not on added Li were washed three times with ice-cold 5% mannitol (without enzymes) and incubated at 4 °C to simulate sorting conditions, as represented in the schematic (**g**). Li levels determined in the supernatant (**h**) and in isolated protoplasts (**i**) right after the last washing step (time point 0) or after 15 and 30 min of incubation at 4 °C. Li content in the same number of protoplasts sorted within 15 min or 30 min after isolation (**j**). Bars and symbols show means ± SD ($n = 4$ biological replicates in **h**, **i** and $n = 3$ biological replicates in **j**). In **a**, **b–f**, **h–j**, *P*-values were determined by two-tailed unpaired Student's *t* test. Source data are provided as a Source Data file.

ICP-MS system) in the reagents used for protoplast isolation and sorting (Supplementary Table 1). Cultivation of *A. thaliana* seedlings on Li-containing solid media for 5 days increased Li levels to 16.5 µg g⁻¹ FW (Supplementary Fig. 4d). Roots were then digested with the modified cell isolation solution for 90 min, and the isolated protoplasts washed as described above with ice-cold 5% mannitol. Compared to pure mannitol or the supernatant containing mock protoplasts, Li levels were ~2.3 times higher in the solution containing Li-enriched protoplasts and remained relatively stable for 15 min while increasing approx. 20% after 30 min of incubation at 4 °C (Fig. 1g–h). This result indicates that Li is exported or released, e.g. when protoplasts burst. In pelleted cells, Li concentrations did not change significantly during incubation at 4 °C (Fig. 1i). We then quantified Li in

100,000 protoplasts sorted within different time intervals after isolation. Although Li contents slightly decreased when sorting was prolonged to 30 min, the decrease was not significant (Fig. 1j). Therefore, we restricted sorting to 30 min after protoplast isolation.

As the most commonly used FACS sheath fluids contain significant levels of various elements (e.g., derived from PBS buffer), we tested the possibility to replace these solutions with a single salt. In initial tests using a BD FACSAria IIu (BD Biosciences) cell sorter, different concentrations of NaCl (Sigma, 99.98%) were tested. A similar concentration of another salt (LiCl, Sigma, ≥99.99%) was also effective on a BD Influx™ (BD Biosciences) cell sorter whereas 2.5 mM LiCl was the best concentration when using a MoFlo XDP (Beckman Coulter) sorter. We compared the elemental levels of sorted solutions

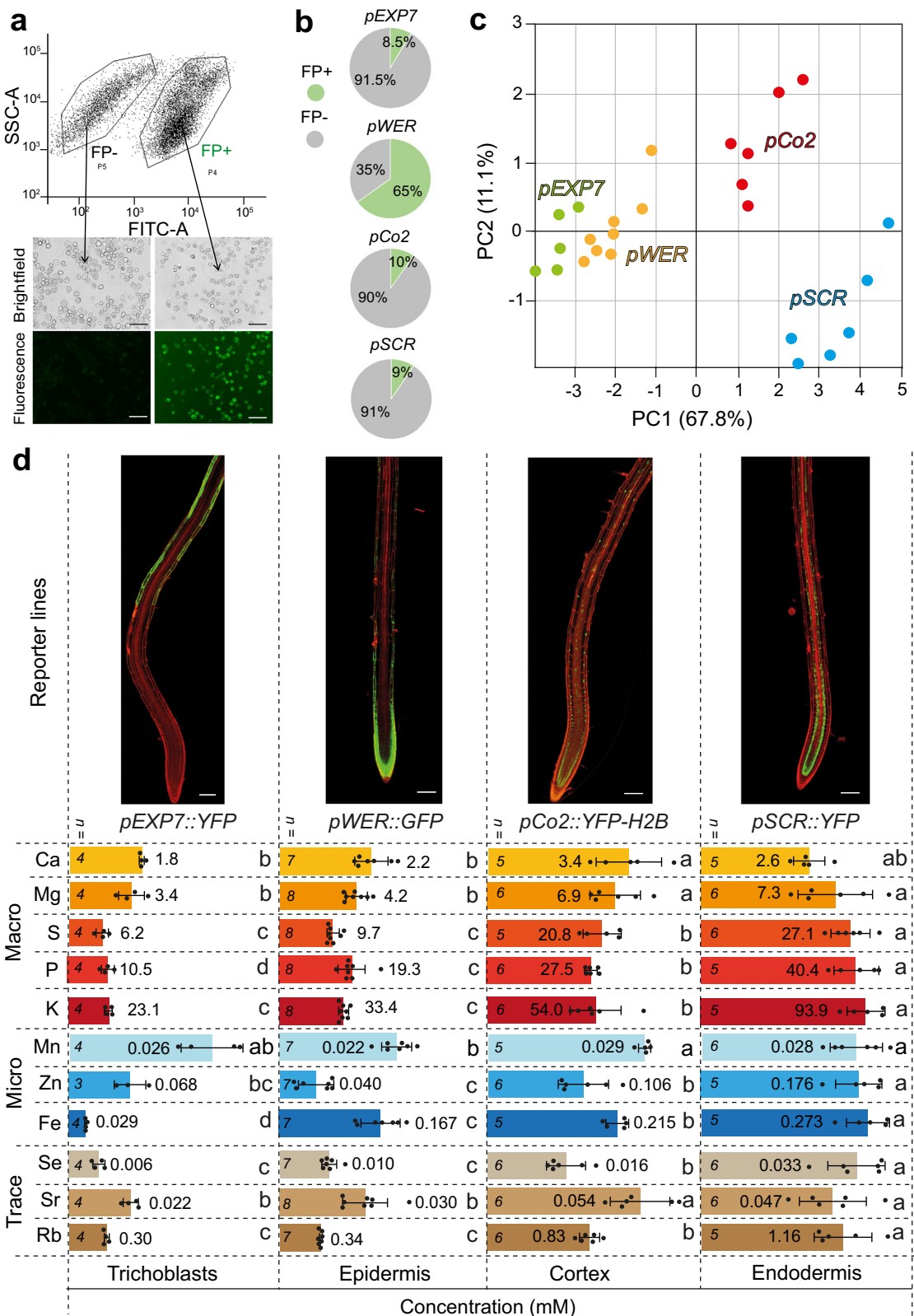

containing different numbers of protoplasts or only FACS sheath fluid (5 mM NaCl) passed through a BD Influx™ (BD Biosciences). This analysis revealed that element contents must be corrected for background contamination if a contaminant-free cell sorter is not available (Supplementary Fig. 5). Thus, we subtracted significant contaminations detected in FACS sheath fluid (without samples) collected regularly during sorting from the levels detected in solutions containing sorted

protoplasts. Furthermore, we found that in our equipment at least 50,000 protoplasts are required for the reliable detection of up to 11 elements of interest (Supplementary Fig. 5).

**Cell type-specific mapping of ion distribution in roots**
After setting up reliable conditions for protoplast isolation and sorting, we used our method to investigate elemental concentrations in four

**Fig. 2 | Cell type-specific element distribution in *A. thaliana* roots.**
**a**, **b** Fluorescence-activated cell sorting of protoplasts prior to ICP-MS. Example of fluorescence intensity versus side-scatter light gating of intact protoplasts from the *pWER::GFP* reporter line used to sort fluorescent protein-positive (FP+) and -negative (FP−) protoplasts (**a**) and sorting efficiencies of different cell types expressed as percentage of all gated events (**b**). Images are representative of experiments repeated at least two times. Scale bars in **a**, 50 μm. **c** Principal component analysis plot illustrating the separation of different cell types according to their ionome. The lines *pEXP7::YFP*, *pWER::GFP*, *pCo2::YFP-H2B* and *pSCR::YFP* were used to sort protoplasts representing trichoblasts, epidermis, cortex and

endodermis, respectively. **d** Concentrations of the indicated elements in four root cell types. Concentrations per unit tissue water (mM) were calculated according to the average volume of each sorted cell type. Average values are shown at element-specific scales. Seedlings were grown for 5 days on solid half-strength MS agar media supplemented with subtoxic levels of Sr, Se, and Rb. Bars represent means ± SD. The number (*n*) of independent replicates analyzed is indicated. Different letters within rows indicate significant differences among means according to one-way ANOVA followed by Tukey's test at *P* < 0.05. Scale bars, 100 μm. Source data are provided as a Source Data file.

different cell types using *A. thaliana* reporter lines described previously[37–39]. Accuracy of sorting was inspected with fluorescence microcopy and the proportion of sorted FP-positive protoplasts determined for each reporter line (Fig. 2a, b). At the whole-root level, the ionome determined with HR-ICP-MS of the reporter lines *pWER::GFP*, *pEXP7::YFP*, *pCo2::YFP-H2B* and *pSCR::YFP* did not differ substantially (Supplementary Table 2). In contrast, ionome profiling of sorted protoplasts indicated that the element composition of cells from the outermost layers (*pEXP7* and *pWER*) separated from those in inner root layers (*pCo2* and *pSCR*) along principal component 1, which explained 67.8% of the variation (Fig. 2c). As the number of sorted protoplasts collected for ICP-MS is known but their mass cannot be directly measured, concentrations were normalized to an estimated protoplast volume based on the diameter of fluorescent protein (FP)-expressing protoplasts (Supplementary Fig. 6). We obtained millimolar concentrations of all determined macroelements with superior abundance of K over P and S (Fig. 2d), which closely agrees with the quantitative dimensions determined by LA-ICP-MS[19] or quantitative x-ray measurements[40]. As expected, Mn, Zn and Fe were in the micromolar range (Fig. 2d). While most mineral element concentrations were quite similar between trichoblasts and epidermal cells, they increased toward the cortex and even further toward the endodermis, reflecting superior nutrient accumulation in the cortex and endodermis. With regard to K, the obtained results are in line with the distribution recently detected with a K⁺-sensitive, genetically encoded FRET-based sensor, which revealed a strong radial gradient of K with lowest concentrations at the epidermis and increasing toward the vascular tissue[41]. Furthermore, the radial distribution of Rb and Se went along with that of K and S, respectively (Fig. 2d).

As a mean to assess element distribution in different cell types without relying on estimated protoplast volumes, we normalized element concentrations in fluorescent protein (FP)-expressing protoplasts against a reference population of FP-negative cells sorted from the same sample. In an independent experiment with three cell reporters, we observed higher relative levels for all elements in *pCo2*- and *pSCR*-expressing than in *pWER*-expressing protoplasts (Fig. 3a). Only Mn made an exception, as it accumulated to a very similar extent in all three cell types. These results reinforced the existence of an inward-pointing concentration gradient for most elements under non-stressed conditions.

**Disturbed metal loading into the xylem results in cell type-specific changes of the root ionome**

In order to reach distant organs and tissues, mineral elements move radially across different cell layers in the root before they are loaded into xylem vessels via specialized ion transporters for long-distance translocation[42]. Using the new approach, we investigated how the distribution of elements changes when this process is disturbed. Firstly, we assessed the consequence of knocking out Heavy Metal ATPase 4 (HMA4), a plasma membrane protein involved in Zn and Cd loading from the pericycle into the xylem[43,44]. We introgressed a *pSCR::GFP* cell-type reporter into the genetic background of the *hma4* mutant. Using FACS in combination with a quadrupole ICP-MS (Q ICP-MS) system, we could detect a significant increase in endodermal Zn

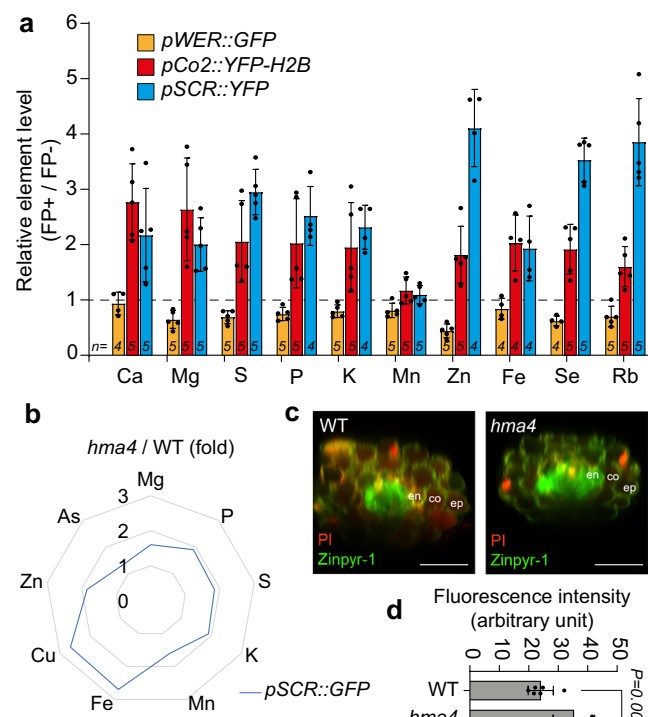

**Fig. 3 | Relative element distributions and the effect of HMA4 mutation in endodermal accumulation of specific elements. a** Independent experiment validating cell type-specific ion distribution. Seedlings of the indicated reporter lines were grown for 5 days on solid half-strength MS agar media supplemented with subtoxic levels of Se and Rb. Elemental contents of 100,000 isolated fluorescent protein-positive (FP+) protoplasts were normalized against non-FP expressing (FP−) protoplasts isolated from the same cell reporter. A value of 1 indicates that the element level of FP+ protoplasts was equivalent to that in reference FP− cells. Bars represent means ± SD. The number (*n*) of independent replicates analyzed is indicated. **b** Radar plots showing multi-element fold changes in endodermal cells (*pSCR::GFP*) of *hma4* roots in relation to wild type (WT). **c**, **d** Radial distribution of Zn determined with the Zn-sensitive fluorescent probe Zinpyr-1 in WT roots and *hma4* mutant roots. Representative images of stained roots (**c**) and quantification of Zinpyr-1 signal in endodermal cells of WT and *hma4* (**d**). Images are representative of experiments repeated at least two times. Scale bars, 50 μm. ep epidermis, co cortex, en endodermis. Bars represent means ± SD (*n* = 6 and 5 biological replicates for WT and *hma4*, respectively). *P*-values were determined by two-tailed unpaired Student's *t* test. Source data are provided as a Source Data file.

concentration in *hma4* mutant (Fig. 3b). This is in accordance with increased signals of Zinpyr-1 – a Zn-sensitive fluorophore – detected in endodermal cells of *hma4* mutant by confocal microscopy (Fig. 3c, d). Interestingly, *HMA4* disruption resulted in a more widespread change in the ionomic profile, as the levels of Fe and Cu were increased even more dramatically in endodermal cells of *hma4* plants than the levels of Zn (Fig. 3b).

Next, we assessed the consequence of disruption of Ferric Reductase Defective 3 (FRD3), a citrate efflux transporter expressed in

the vascular tissue[45]. Previously, it has been shown that *frd3* plants overaccumulate Fe not only in the vascular tissue but also in endodermal cells of roots[8]. However, it remained unclear if the distribution of other metals is also affected if citrate loading into the xylem is disturbed. Whole-root analysis confirmed that disruption of FRD3 causes 2- to 3-fold higher accumulation of Fe, Mn and Zn but not of Ca. Interestingly, with our FACS-HR-ICP-MS method we detected, besides Fe, also higher levels of Zn, Mn and Ca in *pSCR::YFP*-expressing endodermal cells isolated from *frd3* roots (Supplementary Fig. 7 a–e). Besides Fe, also Mn, Zn and to a lower extent even Ca can be transported in the xylem in the form of citrate complexes[46], which will depend ultimately on FRD3. Mn was just slightly increased in the *pWER::GFP*-expressing epidermal cells. The specific increase of Fe in endodermal but not in epidermal cells was confirmed by histochemical Fe visualization with Perls stain (Supplementary Fig. 7f and ref. 8). Altogether, cell type-specific metal mapping reveals that impaired HMA4 and FRD3 function in the vascular tissue impacts the radial distribution of more elements than previously detected with single element detection approaches.

### Cell type-specific vacuolar loading of Mn in roots determines plant tolerance to Mn-induced Fe deficiency

One consequence of the poor selectivity of IRT1 toward different divalent metals is that a series of vacuolar metal transporters are induced in response to Fe deficiency to sequester the excess of non-Fe metals brought in by IRT1[47–50]. Interestingly, many of these vacuolar transporters involved with metal sequestration in roots are expressed in the rhizodermis and cortex[47–50], suggesting that metal detoxification occurs mainly in outer cell layers. To address this possibility, we compared metal levels in *pWER::GFP*- and *pEXP7::YFP*-expressing epidermal cells and trichoblasts with those in *pSCR::YFP*-expressing endodermal cells of Fe-deficient roots. We found that Fe contents were significantly decreased in epidermal cells of Fe-deficient roots while the amounts of Mn, Zn and copper (Cu) increased (Fig. 4a–d). Although root endodermal cells of Fe-deficient plants retained more Fe than epidermal cells, they had comparatively less Zn and Cu and similar contents of Mn. Interestingly, trichoblasts of Fe-deficient roots contained intermediate contents of Fe, Zn and Cu but accumulated significantly more Mn than the other assessed cell types (Fig. 4a–d). This enrichment was specific to YFP-positive protoplasts of the *pEXP7::YFP* reporter line (Supplementary Fig. 8a), further supporting a role for trichoblasts in the accumulation of excess Mn in roots.

In *A. thaliana* roots, the CATION DIFFUSION FACILITATOR (CDF)-type transporter METAL TOLERANCE PROTEIN 8 (MTP8) plays a major role in vacuolar sequestration of Mn[50,51]. To investigate the relevance of cell-type specificity in efficient Mn sequestration along the radial axis of roots, we reintroduced MTP8 either in trichoblasts or in endodermal cells of the *mtp8-1* mutant. The activity of the selected promoters was not significantly affected by Fe-limiting conditions and independent transgenic lines with comparable *MTP8* expression levels were used (Fig. 4e and Supplementary Fig. 8b, c). When plants were grown under Fe-limiting conditions in the presence of Mn, expression of *MTP8* in trichoblasts of *mtp8-1* roots was sufficient to prevent the appearance of Mn-induced Fe-deficiency symptoms in leaves (Fig. 4f). However, complementation was not efficient when MTP8 activity was reconstituted only in endodermal cells. Analysis of Mn levels in roots and shoots revealed that *pEXP7*-driven *MTP8* expression in trichoblasts of *mtp8-1* plants increased Mn retention in roots and decreased Mn accumulation in shoots (Fig. 4g and Supplementary Fig. 8d). The effect was more pronounced under Fe-limiting conditions, when MTP8-dependent Mn accumulation in roots was strongly increased.

To further investigate the importance of trichoblasts in Mn sequestration, we used L-kynurenine (L-kyn), an auxin biosynthesis inhibitor that also suppresses root hair formation[52]. Supply of the inhibitor in the growth medium did not affect root growth, trichoblast specification or the longitudinal length of *pEXP7::YFP*-expressing epidermal cells but inhibited hair elongation (Supplementary Fig. 9a–d). Specifically under Fe-depleted conditions, L-kyn significantly decreased the size of *pEXP7::YFP*-expressing protoplasts, without negatively affecting the expression of *EXP7* or Fe deficiency-induced genes, including *MTP8* (Supplementary Fig. 9e, f). This suggested that the concentration of L-kyn used in these experiments efficiently inhibited the elongation of root hairs without significantly inhibiting Fe responses and overall root growth. Interestingly, FACS-ICP-MS revealed that the inhibitor significantly decreased the Mn content of sorted *pEXP7::YFP*-expressing protoplasts (Fig. 4h). We then investigated the consequence of impaired root hair elongation in root-to-shoot Mn partitioning. Under Fe-limiting conditions, the efficiency of *pEXP7*-driven *MTP8* expression in *mtp8-1* to retain Mn in roots was significantly decreased when root hair elongation was inhibited by L-kyn (Fig. 4i, j). Taken together, our results highlight the importance of cell size and the "topographical" location of metal sequestration mechanisms in roots for plant tolerance to metal toxicity.

## Discussion

Cell-type-specific determination of mineral elements in roots is crucial to understand mechanisms underlying selective nutrient enrichment or partitioning, metal detoxification and mineral element interactions in plants. In this study, we developed a new quantitative method for cell type-specific ionomics, and use it to interrogate the radial distribution of elements in roots, in particular when individual steps in metal radial transport are perturbed. In order to establish interference-free FACS sorting, several modifications in existing methods for protoplast isolation and sorting were necessary. With these modifications, we developed a procedure that enables the isolation of a sufficient number of protoplasts within max. 90 min of cell-wall digestion and that prevents significant leakage of ions during the time required for sorting (Fig. 1g–j). Protoplasts isolated with our method can be used for elemental analysis directly or after sorting. Even without sorting, the analysis of whole-roots and protoplasts from wild-type or mutants plants cultivated under different nutrient availabilities already revealed dynamic changes in the accumulation of different elements (Fig. 1b–f and Supplementary Fig. 3), highlighting the importance of estimating the element fraction immobilized in cell walls. We could also demonstrate that the presented methodology is suitable not only when using the highly specialized double focusing magnetic sector field HR-ICP-MS but also more common and inexpensive Q ICP-MS systems (Fig. 3b).

To obtain estimated concentrations of various elements in sorted cell-type populations, average protoplast size from non-uniform populations were considered. The element concentrations obtained here (Fig. 2d) lie in the range of measurements obtained with ion-selective microelectrodes[53], x-ray quantification[40] or laser capture microdissection coupled with ICP-MS[19]. Independently of whether results were normalized to estimated protoplast volumes or referred against concentrations detected in FP-negative protoplasts from the same root samples, we identified the existence of gradients for most elements between the outer cells toward the inner layers and confirmed that Mn does not accumulate in radial direction (Fig. 2d, Fig. 3a). In the case of K, this finding agrees with the results obtained with a FRET-based K reporter[41]. Although it remains unclear how cell-to-cell gradients are established, heterogenic element distribution likely originates from i) the differential expression of plasma-membrane and tonoplastic ion transporters, and ii) the cell type-specific abundance of plasmodesmata determining the efficiency of symplastic transfer between neighboring cell layers. A quick transit of elements through the rhizodermal layer could potentially prevent that nutrient importers located at the plasma membrane of these cells are inhibited by the overaccumulation of their substrate in the cytosol or vacuoles. By combining our FACS-ICP-MS method with recently

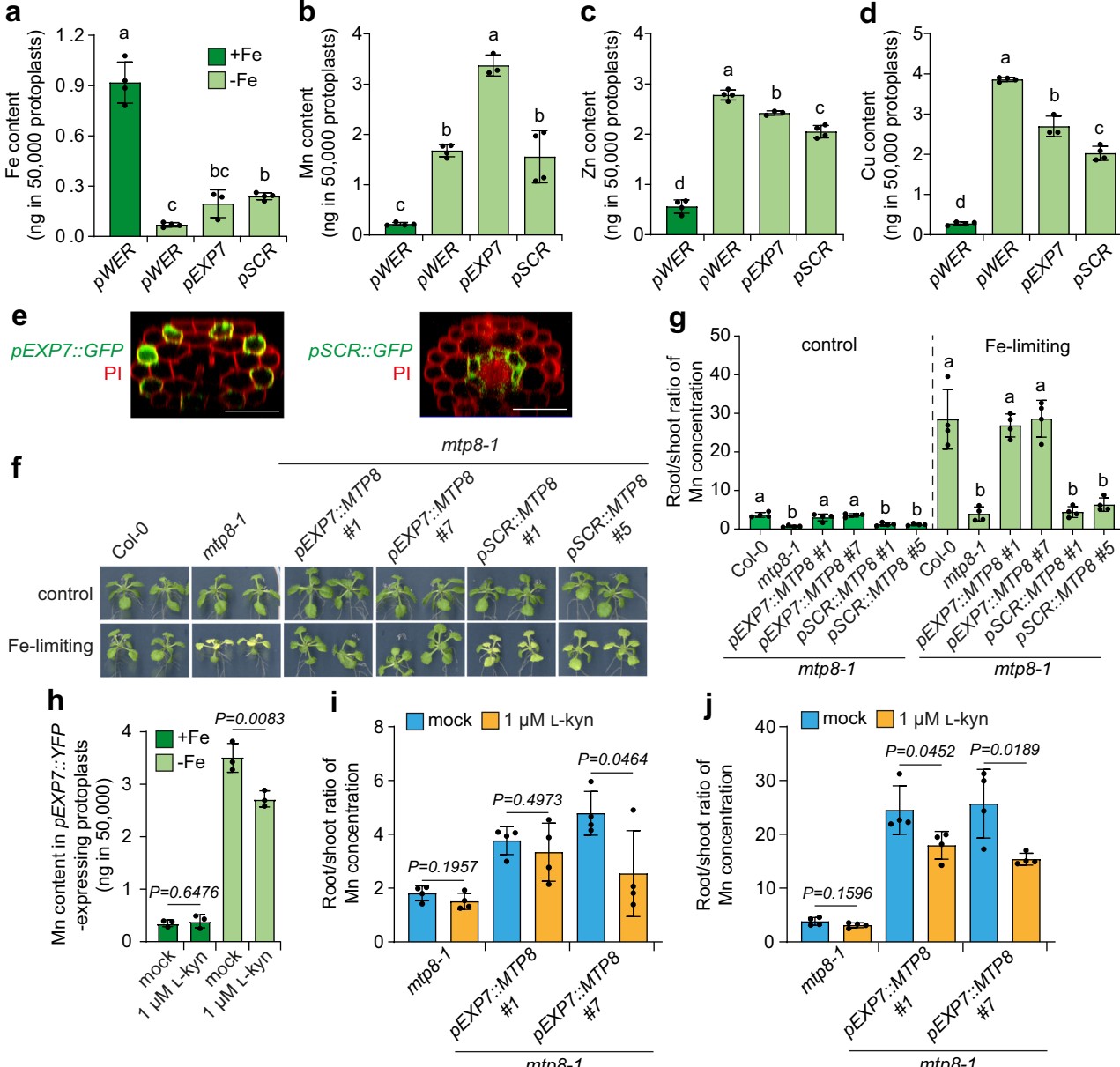

**Fig. 4 | Role of trichoblasts in Mn detoxification. a–d** Increased Mn accumulation in *pEXP7*-expressing trichoblast cells of roots grown under low Fe. Cell type-specific contents of Fe (**a**), Mn (**b**), Zn (**c**), and Cu (**d**) in sorted protoplasts. *pEXP7*, *pWER* and *pSCR* are expressed in trichoblasts, epidermis and endodermis, respectively. Seedlings were grown for 5 days on solid half-strength MS agar media containing 100 μM Fe (+Fe) or without added Fe (−Fe). Bars are means ± SD (n = 4, 4, 3, and 4, respectively in **a**; n = 4, 4, 3, and 4, respectively in **b**; n = 4, 4, 3, and 4, respectively in **c**; and n = 4, 4, 3, and 4, respectively in **d**). Different letters indicate significant differences among means according to one-way ANOVA followed by Tukey's test at P < 0.05. **e** Localization of the promoters cloned to drive cell type-specific *MTP8* expression. Shown are reconstructed confocal Z-stacks from representative lines grown on Fe-limiting conditions. Images are representative of experiments repeated at least two times. Scale bars = 50 μm. **f, g** Expression of the vacuolar Mn transporter *MTP8* in trichoblasts efficiently retains Mn in Fe-deficient roots. Appearance of Mn-induced Fe-deficiency symptoms in shoots (**f**) and ratio of Mn concentrations between roots and shoots (**g**). Root and shoot Mn concentrations are shown in Supplementary Fig. 8. Ten-day-old seedlings were transferred to solid half-strength MS agar media containing 30 μM Fe-EDTA and 80 μM Mn at pH 5.5 (control) or pH 6.7 (Fe-limiting) and grown for 8 days. Data are means ± SD (n = 4 independent biological replicates containing 5 plants each). Different letters within each Fe availability condition indicate significant differences among means according to one-way ANOVA followed by Tukey's test at P < 0.05. **h** Mn content in sorted *pEXP7::YFP*-expressing protoplasts isolated from plants grown for 5 days on solid half-strength MS agar media containing 100 μM Fe (+Fe) or without added Fe (−Fe) and 1 μM of the inhibitor L-kynurenine (L-kyn) or only the solvent (mock). **i, j** Effect of inhibiting root hair elongation on the ratio of Mn concentrations between roots and shoots of plants grown under sufficient (**i**) or limited Fe availability (**j**). Ten-day-old seedlings were transferred to solid half-strength MS agar media containing 30 μM Fe-EDTA and 80 μM Mn at pH 5.5 (**i**) or pH 6.7 (**j**) supplemented with L-kyn or the solvent DMSO (mock) and grown for 8 days. Data are means ± SD (n = 4 independent biological replicates containing 5 plants each). In **h–j**, P-values were determined by two-tailed unpaired Student's t test. Source data are provided as a Source Data file.

developed cell type-specific CRISPR mutagenesis[54,55] and/or inducible systems that can spatially constrict plasmodesmata[56], it will be possible to test these hypotheses.

Despite opening new analytical possibilities, our FACS-ICP-MS method has some limitations. As our method determines the ionome of protoplasts, it detects concentration changes primarily associated with intracellular pools. It is noteworthy that, since protoplasts from different cell types and originating from different root zones differ in size, by taking an average volume of a pooled population of sorted protoplasts, our method is limited in absolute quantification. Nevertheless, it allows for relative comparison of the ionome of specific cell types in response to genetic perturbations or external cues (Figs. 3a, b and 4a–d, and Supplementary Fig. 7e). As we intended to also include elements of low abundance, we could only analyze cell types of which abundant protoplasts could be isolated and sorted within short times in order to avoid significant changes in elemental concentrations. Thus, we were not yet able to achieve the same spatial resolution as reported previously for FACS-transcriptomics[21]. In future, the method can be further improved by using reporter lines with higher cell type- and root zone-specificity and by decreasing the number of protoplasts required for reliable ICP-MS quantification. To access cell types comprised of a small number of cells per root (e.g., columella cells) or that are more difficult to protoplast (e.g., pericycle cells), contaminant-free laboratory space, dedicated instrumentation as well as ultraclean reagents and labware will be necessary. Moreover, detection limits could be further improved by employing single-particle analysis with time-of-flight ICP-MS or by equipping common ICP-MS devices with more accurate sample introduction systems that allow maximum transfer efficiencies[57]. With such improvements, more precise analysis along the longitudinal axis can be performed.

As a proof-of-concept, we used our method to investigate cell type-specific ionomic changes in response to limited Fe availability. We could demonstrate that Mn accumulation increases in Fe-deficient epidermal cells (Fig. 4b), probably as a result of increased Mn vacuolar loading by MTP8. Furthermore, we detected more Mn in hair-forming epidermal cells than in *pWER*-expressing epidermal cells or *pSCR*-expressing endodermal cells. This strong cell-type dependency matches well the cell type-specific expression of *MTP8*[50] and showcases how the nutrient status of a plant alters the cell type-specific ionome of its roots. Since in Fe-deficient roots *IRT1* is mainly expressed in trichoblasts[33], our results highlight the critical importance of immediate vacuolar sequestration of Mn at the site of uptake for plant tolerance to Fe deficiency-induced Mn toxicity. This was confirmed by demonstrating that Mn can be efficiently retained in roots by confining *MTP8* expression in trichoblasts but not in endodermal cells (Fig. 4g). Root hair formation is induced by many nutrient deficiencies, including Fe deficiency[58,59]. This response is thought to enlarge the absorptive surface of roots but may also increase the capacity for vacuolar storage of non-Fe metals taken up by IRT1. Our results show that increased cell size allows the accumulation of more Mn in trichoblast of Fe-deficient plants and that inhibition of root hair elongation significantly decreases MTP8-mediated Mn sequestration in roots (Fig. 4h–j). These findings demonstrate how FACS-ICP-MS can help to uncover novel mechanisms underlying elemental distribution in plants. We expect that this method will help to determine how individual transporters and their expression domains shape the ionome in different root tissues and cell layers, and finally determine nutrient partitioning to shoots. Furthermore, the possibility to combine our method with transcriptomics and to develop it further toward single cell ICP-MS paves the way to investigate transcriptome-ionome networks at very high spatial resolution. This knowledge is critical to understand and manipulate transport pathways in order to increase nutrient use efficiency while simultaneously preventing accumulation of toxic elements in aboveground tissues.

## Methods

### Plant materials and growth conditions

The *A. thaliana* reporter lines used in this study were *pEXP7::YFP*[38], *pWER::GFP*[37], *pCo2::YFP-H2B*[39], *pSCR::YFP*[38], and *pSCR::GFP*[20,60]. Seeds were obtained from Kenneth D. Birnbaum (University of New York), Philip N. Benfey (Duke University, Durham, US) or NASC.

Seeds were surface sterilized in 70% (v/v) ethanol and 0.05% (v/v) Triton X-100 for 15 min and washed twice with 96% (v/v) ethanol. Sterilized seeds were sown onto solid medium containing 1% (w/v) Difco agar (Becton Dickinson), 0.5% (w/v) sucrose, 1 mM MES pH 5.5 and half-strength Murashige and Skoog (MS) medium (Duchefa Biochemie, catalog No. M0021) supplemented with subtoxic concentrations of Sr (100 μM as $SrCl_2$), Se (2 μM as $NaSeO_4$) and Rb (200 μM as RbCl). For the Li enrichment experiment, 5 mM Li was additionally supplied as LiCl. Up to 900 seeds were sown in three rows per square petri dish. After sowing, plates were kept at 4 °C for 2 days and then transferred to growth cabinets under the following regime: 10/14 h light/dark; light intensity 120 μmol m$^{-2}$ s$^{-1}$ (fluorescent lamps); temperature 22 °C/18 °C (light/dark). Following a previously established protocol[28], P-depleted medium contained 1% (w/v) Difco agar (Becton Dickinson), 0.5% (w/v) sucrose, 1 mM MES pH 5.5 and half-strength MS salts with $KH_2PO_4$ replaced by KCl, whereas in the K-depleted medium $KNO_3$ and $CaCl_2$ were replaced by $Ca(NO_3)_2$ and $KH_2PO_4$ by $NaH_2PO_4$. Fe deficiency for cell type-specific quantification of metals was induced by growing plants on half-strength MS-based solid media as described above but without added Fe-EDTA. To test *mtp8-1* complementation, plants were grown on half-strength MS-based solid media containing 30 μM $FeCl_3$ and 80 μM $MnCl_2.4H_2O$ with pH values adjusted to 5.5 (control) or 6.7 (Fe-limiting) with 10 mM MES buffer. Root hair elongation was inhibited by supplying 1 μM L-kynurenine (CAS Number 2922-83-0, Sigma) to half-strength MS-based solid media. For the experiment shown in Fig. 3b, sterilized seeds were sown onto the sterilized 100 μm Nitex mesh that was overlaid onto the surface of solid medium containing 1% (w/v) agar (Fisher Scientific), 0.1% sucrose (w/v) and half-strength Murashige and Skoog Basal Salt Mixture (MS) (Sigma-Aldrich, catalog no. M5524) at pH 5.7, supplemented with subtoxic concentration of arsenic (As; 0.5 μM $NaAsO_2$). The seeds were evenly sown in three rows with up to 1200 seeds per square petri dish. After sowing, plates were kept at 4 °C for 2 days and then transferred to the growth room and cultivated under following regime: 16/8 h light/dark, light intensity 100 μmol m$^{-2}$ s$^{-1}$, temperature 22 °C/19 °C (light/dark).

### Protoplast isolation with modified protocol

To isolate root protoplasts, the standard protoplasting solution[61] was replaced by a modified solution without KCl, $MgCl_2$, $CaCl_2$ and bovine serum albumin. The modified solution contained only 5% mannitol (w/v) buffered to pH 5.6 with 5 mM MES. This solution was used to dissolve 10 mg mL$^{-1}$ cellulase "Onozuka" RS from *Trichoderma viride* (16420, Serva®) and 1 mg mL$^{-1}$ macerozyme R-10 from *Rhizopus* sp. (28302, Serva®). Protoplast isolation was also efficient by using the modified solution containing only 7% mannitol (w/v) and 15–20 mg mL$^{-1}$ cellulase from *Trichoderma viride* (Sigma Aldrich) and 3 mg mL$^{-1}$ pectinase from *Rhizopus* sp. (Sigma Aldrich). Roots of five-day-old plants were sliced off with a razor blade. Roots were then rinsed three times with MQ water, cut into small pieces with a razor blade, and then transferred to freshly prepared protoplasting solution. Samples were incubated at 22 °C for 90 min on a bench-top orbital shaker set at 80 r.p.m. After 90 min incubation in darkness, the solution containing the tissue and enzymes was aspirated with a 5mL-pipette and passed through a 40-μm cell strainer. Then, a clean pipette was used to gently aspirate the cell suspension and dispense it into cold 15-mL-screw-top tubes to a maximum of 10 mL per tube. Cell suspension was centrifuged at 500 *g* for 5 min at 4 °C to collect the protoplasts. After centrifugation, the supernatant was carefully

removed without disturbing the protoplast pellet. Protoplasts were then gently resuspended in ice-cold 5% mannitol (no enzymes, no MES buffer), starting with a wide-bore (a cut-off 1-mL tip) pipette tip and then using a smaller bore (uncut 1-mL tip) pipette tip. The suspension was then transferred to a new 15-mL-screw-top tube and centrifuged at 500 $g$ for 1 min at 4 °C. After removing the supernatant as described above, the procedure was repeated one more time so that the pellet was washed two times with 5% mannitol. In the final step, protoplasts were resuspended in 0.5–1.0 mL ice-cold 5% mannitol and sorted within max. 30 min. To compare elemental levels in epidermis and endodermis of WT and *frd3* roots, root factions of approximately 5 mm long were collected 1 mm away from the root tip to enrich samples with fractions from the differentiation zone.

## Fluorescence-activated cell sorting

Protoplasts were sorted using either a FACSAria IIu (BD Biosciences) or a BD Influx™ (BD Biosciences) cell sorter. To maintain the integrity of the protoplasts during the sorting, sample tubes were cooled (approx. 5 °C) and protoplasts sorted with a 100-μm nozzle. The sheath pressure was set to 20 psi and as sheath fluid 5 mM solution of NaCl in MQ water was used. For results shown in Fig. 3b, LiCl was used instead of NaCl. The sheath fluid composition and concentration were optimized to reduce the matrix effect and the background signals of the elements to be measured by ICP-MS. Intact protoplasts were first identified in a dotplot displaying forward-scatter light versus side-scatter light. The corresponding sorting gates to separate the fluorescence-positive and -negative protoplasts were defined in a dotplot showing the GFP/YFP fluorescence versus side-scatter light (Supplementary Fig. 10). To define the FP-positive gate boundaries properly, the FP-negative protoplast sample was prepared for each sorting experiment. The successful sorting of both populations was controlled using an Axiophot 2 fluorescence microscope (Zeiss) equipped with a cooled CCD camera (ORCA ER, Hamamatsu). To estimate contaminations introduced by the cell sorter, at regular intervals the devices were switched to test mode and sheath fluid was collected at the same volume as for samples.

## Element analysis with sector field high-resolution ICP-MS

Depending on the experiment, 50,000 or 100,000 protoplasts were necessary to run multi-element analysis. As sorted protoplasts were digested prior to ICP-MS analysis, it was not necessary to maintain their viability after sorting. Two independent procedures were tested for digestion. In initial experiments, protoplasts were sorted into 1.5-mL microcentrifuge tubes. The content (150–200 μl) was then transferred to ultra-clean polytetrafluoroethylene tubes for digestion with concentrated nitric acid (Suprapur®) in a pressurized system (UltraCLAVE IV, MLS). De-ionized distilled water from the Milli-Q® Reference System (Merck, Germany) was used to adjust the volume to 1.5 mL. In later experiments, protoplasts were sorted directly into 15-mL-screw-top tubes and acid-digested (10% nitric acid) under a fume hood. For this procedure, it was critical to wash all tubes and pipette tips with a 10% nitric acid solution prior to use. Element standards were prepared from certified reference single standards from CPI-International (USA). The analysis of all mineral elements was performed with a sector field high-resolution (HR)-ICP-MS (ELEMENT 2™, Thermo Scientific™, Germany) with software Element version 3.1.0.236 and the parameters indicated in Supplementary Table 3. Blanks (only 5% mannitol) and FACS sheath fluid were digested and analyzed the same way as protoplast samples. Due to low concentrations in protoplasts and significant background contamination, this ICP-MS system was not able to reliably quantify the micronutrients B, Cu, cobalt (Co), nickel (Ni) and molybdenum (Mo) even in 100,000 sorted protoplasts. Furthermore, whenever NaCl was used as the sheath liquid during sorting, sodium (Na) and chlorine (Cl) were not assessed.

For the elemental analysis of whole shoots and roots, samples were first dried at 65 °C and digested in concentrated $HNO_3$ in polytetrafluoroethylene tubes under pressurized system (UltraCLAVE IV, MLS) and then analyzed with HR-ICP-MS (ELEMENT 2™, Thermo Scientific™, Germany) as described above.

## Element analysis with quadrupole ICP-MS

Independently of the method development at IPK, a similar approach was developed at the University of Nottingham using Q ICP-MS. In this method, approximately 50,000 protoplasts were directly sorted to acid washed 0.7 mL HPLC PP vials to be subsequently acidified with 67–69% nitric acid (Optima grade, Fisher Scientific) to obtain the final concentration of nitric acid of 3.5%. The samples were then analyzed using a Q ICP-MS (7900, Agilent Technologies, UK) fitted with a collision/reaction cell. The peristatic pump on the ICP-MS was equipped with TYGON tubing with an inner diameter of 0.19 mm (TYGON R3607 0.19 I.D., Agilent Technologies, UK), generating a flow of 40 μL min⁻¹. Multi-element analysis was performed in three different collision/reaction cell gas modes (helium, hydrogen and oxygen). The system was rinsed with 3.5% $HNO_3$ between every sample. Blanks (3.5% nitric acid and 5% mannitol) and FACS sheath fluid were prepared for ICP-MS analysis in the exact same manner as protoplast samples. External calibration was made using a custom-made, non-equimolar multi-element standard (P/N 4400-ICP-MSCS, CPI International, Santa Rosa, USA) which corresponds to concentrations typically found in plants.

## Staining of Fe and Zn and confocal microscopy

Histochemical staining of Fe was performed as described previously[62] by infiltrating intact roots with equal volumes of 4% (v/v) HCl and 4% (w/v) K-ferrocyanide (Perls stain solution) for 15 min followed by 30 min incubation at room temperature. To visualize Zn distribution in roots, we incubated roots with 10 μM of the Zn-sensitive fluorophore Zinpyr-1 (Chemodex, catalog No. Z0001) for 3 h in darkness, as described by ref. 9. Zinpyr-1 and reporter lines were imaged with a Zeiss LSM 780 (Carl Zeiss) or Leica SP8 (Leica Microsystems GmbH) confocal laser scanning microscope. To visualize cell boundaries, roots were stained with 10 μM propidium iodide (Sigma-Aldrich). Laser line excitations and emissions were as follows: GFP (488 nm; 505-550 nm), YFP (514 nm; 518-564 nm) and PI (561 nm; 592-617 nm). Tile scans and orthogonal views of Z-stacks were obtained using the ZEN Black (Zen 2.3 SP1 FP1) while fluorescence was quantified with ZEN 2.6 (blue edition) software.

## Introgression of reporters in mutant backgrounds

Fluorescent reporters to mark epidermal and endodermal cells were introduced into the previously described *frd3-1*[63] and *hma4-2*[43] mutants via crossing. Lines expressing the reporters and homozygous for the *frd3-1* mutation were selected in the F2 population by confirming the simultaneous presence of GFP- or YFP-derived signal with the help of a fluorescence microscope (Imager 2, Zeiss) and the expected phenotypical changes associated with *FRD3* mutation (i.e., Fe deficiency symptoms even when grown under sufficient Fe supply and increased Fe-dependent Perls staining in innermost root tissues). The expected increase of Fe and Mn accumulation in roots of introgressed lines carrying the homozygous *frd3-1* allele was confirmed in F3 plants by ICP-MS analysis (Supplementary Fig. 7a–c).

Lines expressing *pSCR::GFP* and homozygous for the *hma4-2* mutation were selected in the F2 population by confirming the simultaneous presence of GFP-derived signal using an epi-fluorescence microscope (Leica CTR5000), 20X magnification with a GFP filter, and polymerase chain reaction (PCR). For DNA extraction, 500 μL of extraction buffer (0.055 M CTAB, 1.8 M NaCl, 0.02 M EDTA, pH 8.0, 0.1 M TRIS, pH 8.0) was added to ground

leaf material and samples were incubated at 60 °C for 25 minutes. After centrifugation (10 min, 13,400 g), 500 µL of chloroform (chloroform/isoamyl alcohol 24:1) was added to the supernatant and samples were vortexed and subsequently centrifuged (10 min, 13,400 g). The supernatant was then mixed with 500 µL of isopropanol and centrifuged (10 min, 13,400 g). Obtained pellet was washed with 70% ethanol and then resuspend in 50 µL of the sterilized water and stored at −20 °C. One microliter of this solution was used as template for PCR. To detect T-DNA insert and WT copy of *HMA4*, the oligos listed in Supplementary Table 4 were used.

### Genetic complementation of *mtp8-1*

The promoter sequences of *EXPANSIN7* (1416 bp; AT1G12560) and *SCARECROW* (1935 bp; AT3G5422), and the full 1896-bp-long coding sequence of MTP8 (AT3G58060) were amplified from genomic DNA with oligos listed in Supplementary Table 4. Promoter sequences were cloned in pGGA000 and the coding sequence of MTP8 in pGGC000, following the GreenGate procedure described in ref. 64. Final constructs were assembled in the pFASTR A-G destination vector[54] containing the *pOLE1::mRuby3* screenable marker[65]. The resulting plasmids were introduced into *Agrobacterium tumefaciens* strain GV3101. To verify in which cell types the cloned promoters of *EXP7* and *SCR* were active, transcriptional fusions with GFP were prepared via GreenGate method using the pGGC025 (3xGFP) module developed by ref. 64. Transformation of *A. thaliana* Col-0 (for promoter::GFP lines) or the *mtp8-1* mutant (for cell type-specific complementation) was performed through the floral dip method[66]. Positive transformants were selected based on the presence of red fluorescence in seeds observed with a fluorescent stereomicroscope (SteREO Discovery.V8, Zeiss, Germany). For cell type-specific *mtp8-1* complementation, at least five independent T2 lines of each transformant were analyzed and the results of two representative lines are shown.

### RNA extraction and real-time quantitative PCR

Total RNA was extracted from the homogenized samples using the NucleoSpin RNA Mini Kit (Machery-Nagel), followed by on-column DNase treatment (QIAGEN), according to the manufacturers' protocols. cDNA was synthesized from 0.5 to 1 µg RNA by reverse transcription using the RevertAid First Strand cDNA synthesis Kit (Thermo Fisher Scientific) and oligo(dT) primer. A 10- or 20-times diluted cDNA sample was then used for quantitative real-time (RT) PCR analysis with the CFX384 Touch Real-Time PCR Detection System (Bio-Rad Laboratories) and the iQ SYBR Green Supermix (Bio-Rad Laboratories), using the listed in Supplementary Table 4. Amplification cycle protocols were as follow: 2 min at 95 °C; 40 cycles of 6 s at 95 °C and 30 s at 60 °C. Melting curves were verified at the end of 40 cycles for confirmation of primer specificities. All reactions were repeated in two technical and three biological replicates. Average Cq values were normalized by ΔΔCT formula against *UBIQUITIN 2* (AT2G36170) or *ACTIN 2* (AT3G18780) expression, as indicated in the legend of the corresponding figures. Values of independent biological replicates are provided in the Source Data file.

### Statistical analysis

To analyze the significant differences among multiple groups, one-way ANOVA followed by Tukey's test at $P < 0.05$ was adopted. The statistical significance between two groups was assessed by two-tailed Student's $t$ test. Statistical tests were performed using SigmaPlot 11.0 and GraphPad Prism 9.3.1 software or the agricolae v.1.3-3 package in R.

### Reporting summary

Further information on research design is available in the Nature Portfolio Reporting Summary linked to this article.

## Data availability

All data generated during this study are included in this published article (and its Supplementary Information File). Source data are provided with this paper.

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

## Acknowledgements

This work was supported by a grant from the Deutsche Forschungsgemeinschaft (Eigene Stelle, HE 8362/1-1) to R.F.H.G., and a grant from The Leverhulme Trust (RPG-2014-058) to D.E.S. We thank Dr. Yudelsy A. Tandron Moya for improving and conducting HR-ICP-MS analyses, Jacqueline Fuge, Annett Bieber, and Lisa Gruber for excellent technical assistance and Dr. Benjamin D. Gruber and Dr. Zhongtao Jia for discussions (Leibniz Institute of Plant Genetics and Crop Plant Research). We also thank Dr. Raif Yuecel, Dr. Attila Bebes and Linda Duncan from Iain Fraser Cytometry Centre at the University of Aberdeen, and Dr. David Onion and Nicola Croxall from Flow Cytometry Facility at the University of Nottingham for the excellent assistance with protoplast sorting experiments, and Dr. Thomas H. Hansen, Dr. Daniel P. Persson, and Dr. Søren Husted (Department of Plant and Environmental Sciences, University of Copenhagen) for the excellent assistance with the Q ICP-MS analysis. We are grateful to Prof. Dr. Kenneth D. Birnbaum (New York University) and Prof. Dr. Philip N. Benfey (Duke University, Durham, US) for sharing reporter lines. Costs for open access publishing were partially funded by the Deutsche Forschungsgemeinschaft (DFG, German Research Foundation, grant 491250510).

## Author contributions

R.F.H.G., N.v.W., and D.E.S designed research; R.F.H.G., P.F., J.F., and Y.G. performed research; R.F.H.G., P.F., J.F., D.E.S, and N.v.W. analyzed data; R.F.H.G. wrote the manuscript with inputs of all authors.

## Funding

## Competing interests

The authors declare no competing interests.
