## [Peer Review File · Nature Communications]

Cell type-specific mapping of ion distribution in *Arabidopsis thaliana* rootsReviewer #1 (Remarks to the Author):

High-resolution cell specific analysis of ion distribution is very interesting and important. Giehl et al developed a measurement that it analyses ionome of different cell populations in roots of *Arabidopsis thaliana* by combination with ICP-MS and fluorescence-activated cell sorting (FACS). Their method provides a way to investigate element transport pathways at cell level in plants. Contaminant control is very important to analyze the plant ionome. They optimized cell wall lysis buffer and conditions to sort protoplasts prior to ICP-MS analysis. With this method, they revealed that most elements exhibit a radial concentration gradient increasing from the rhizodermis to inner cell layers. Ion concentrations of several related mutants were measured, they revealed more elements change than previously detected with single element detection methods. Their results were also proved by different elements detection method, such as fluorescent biosensors. They also found a strong accumulation of manganese in trichoblasts of iron-deficient roots through cell type-specific MTP8 expression for genetic complementation of *mtp8-1*.

But several plant elements concentration under stress conditions can't be measured accurately by ICP-MS-FACS method. Elements, such as Ca and Mg, were enriched in cell wall under -K conditions. Thus, the discussion should include this limitation. They found a strong accumulation of manganese in trichoblasts of iron-deficient roots (Fig. 4). Roots from 5-day-old seedling were analyzed in this study. Root hair has great contribution to ion uptake at this stage. To further support the conclusion, I think a root hair less mutant should be used as a control. It should be determined whether root hairs loss affects accumulation of manganese in trichoblasts.

Minor comment:

Fig.1 b-f check the spelling error "Concentration"

Reviewer #2 (Remarks to the Author):

The manuscript by Giehl et al. described a cell type-specific mapping of element distribution in *Arabidopsis thaliana* roots by the method that combines fluorescence-activated cell sorting and ICP-MS. It seems the accurate and precision determination of the mineral element in protoplasts of *A. thaliana* roots is one of the key steps. In this work, the analysis of the mineral elements was performed with two instruments: a sector field HR-ICP-MS and Q ICP-MS. The authors should provide the detailed information of the parameters of ICP-MS for the determination. The HR-ICP-MS with collision and reaction cell technologies is capable of analyzing most elements in the periodic table from lighter (Li), transition to heavy metals (U) simultaneously at levels of 10 pg/ml with a mass resolution of <1 amu because the matrix interferences are reduced or removed prior to detection. The related information such as isotopes, mass resolution, mass window, sample time, integration window, detection mode, etc. should be provided. However, when using Q ICP-MS, even with collision cell technique, the polyatomic interferences induced by the system and the complex matrix would be very severe especially in the mass range of 31–80, thus the precision determination of ^{56}Fe , ^{57}Fe , ^{64}Zn , ^{66}Zn , ^{67}Zn , ^{76}Se , ^{77}Se , ^{78}Se , ^{80}Se ect. are very difficult especially at very low concentration.

Another difficulty for the cell-specific element analysis is the contaminant-free and integrity isolation of protoplasts from roots. Thus the images showing the integrity of protoplasts should be provided. Fig. 1 h,i were used to estimate the potential element leakage during isolation, "Compared to pure mannitol or the supernatant containing mock protoplasts, Li levels were approx. 2.3 times higher in the solution containing Li-enriched protoplasts", but figure 1h shows the opposite results, furthermore, the result indicated that Li was released from protoplasts. Thus, the obtained data show that all most all the detected elements including the Mg and P were relatively low enrichment in protoplasts compared with in the roots (Supplementary Figure 3), I doubted that it because of the leakage of mineral elements from protoplasts during extraction. In addition, the unit of element concentration is $\mu\text{g/g DW}$, what is the meaning of "DW"?

Collectively, the methodology used in the study is not innovative enough that could not provide high spatial compartmentation of the distribution patterns of important mineral elements in the plants.

Reviewer #3 (Remarks to the Author):

The authors describe the development of a cell-type specific approach to map the ionome in Arabidopsis roots. Optimizing FACS-ICP-MS for this purpose enabled them to generate a multi-element map of ion distribution throughout the layers of root tissue. They uncover a gradient of elemental distribution across the radius of the root. Furthermore, they find alterations of elemental distribution patterns upon disruption of xylem loading through two individual mutant lines. Finally, they found that trichoblast-specific loading of manganese is critical for the stress response to iron deficiency. Overall, this work provides an excellent example of how novel integration of traditional chemical and biological methods can lead to new discoveries and mechanistic understanding. In addition, it has important applications for understanding how plants acquire, store, and transport nutrients and toxins.

The data shown largely supports the conclusions of the manuscript. Overall, the methodology is sound and the methods are sufficiently detailed. Below are the minor suggestions I have for further improving the manuscript:

1) Abstract, line 42: The authors describe how trichoblast Mn sequestration "efficiently retained manganese in roots and prevents toxicity in shoots." The way it is written, it sounds as if the roots need the Mn. My understanding is that this retention is necessary to prevent Mn toxicity in the rest of the plant. If this is correct, I suggest rewording to: "Restricting manganese sequestration in trichoblasts but not in endodermal cells efficiently retained manganese in roots, therefore preventing toxicity in shoots. These results hint to cell type-specific..."

2) The optimized FACS-ICP-MS procedure relies on washing/incubating cells in 5% mannitol and or 5 mM salt, which are compounds frequently used to induce abiotic stress responses in plants (generally at higher concentrations, but protoplasts may experience stress at lower concentrations than intact plants). Do the authors see signs of stress in the cells after FACS? This could be assessed by measuring gene-expression levels in the ABA pathway. There are known links between ABA signaling and iron homeostasis, so if the optimized procedure is causing stress that may influence element levels in the cell, that should be discussed briefly in the manuscript.

3) Page 5, line 163: The logic of this paragraph is difficult to follow. 1) It is a little bit confusing to follow the irt1 mutant experiments vs. the Fe deficiency experiments. I think this section and the discussion would be easier to follow if the term Fe-depleted media was used instead of "Fe deficient plants." 2) I also don't understand – if IRT1 has poor selectivity for Fe²⁺, then is it more selective for Mn and Zn? If so, why doesn't Mn decrease in the irt1 mutant in +Fe conditions?

4) What is your limit of detection with this method (in terms of cell number)? This information would fit into the FACS optimization sections. Right now, it is referenced in the discussion (Line 348-349) but the number is not present in the results section.

5) Line 201: Recommend rewording sentence to say, "...sorted samples if a contaminant-free cell sorter is not available..."

6) It seems like characterizing a stele reporter line would provide further evidence about the enrichment of ions towards the inner root.

7) In the section starting on line 239, the concept of xylem loading is briefly introduced. I think the authors should include a more thorough description of xylem loading, especially in the context of ion transport. I would not expect a biologist unfamiliar with plant biology to understand this section without more information about xylem loading.

8) Line 243: Should it read, "Heavy Metal ATPase 4 (HMA4)"?

9) Line 255 – 258: The hypothesized relationship between citrate and Mn, Zn, and Ca should be explained.

10) MTP8 should be italicized, this is done inconsistently throughout the manuscript.

11) Methods Section, line 380: Why did you use two different pSCR lines?

REVIEWER COMMENTS

We thank the reviewers for appreciating our work and for providing constructive criticisms and suggestions. The manuscript has been substantially revised and we addressed all points raised by the reviewers. We did this by conducting new experiments and by clarifying several issues in the text. With these revisions, we think our manuscript has greatly improved.

Reviewer #1 (Remarks to the Author):

High-resolution cell specific analysis of ion distribution is very interesting and important. Giehl et al developed a measurement that it analyses ionome of different cell populations in roots of *Arabidopsis thaliana* by combination with ICP-MS and fluorescence-activated cell sorting (FACS). Their method provides a way to investigate element transport pathways at cell level in plants. Contaminant control is very important to analyze the plant ionome. They optimized cell wall lysis buffer and conditions to sort protoplasts prior to ICP-MS analysis. With this method, they revealed that most elements exhibit a radial concentration gradient increasing from the rhizodermis to inner cell layers. Ion concentrations of several related mutants were measured, they revealed more elements change than previously detected with single element detection methods. Their results were also proved by different elements detection method, such as fluorescent biosensors. They also found a strong accumulation of manganese in trichoblasts of iron-deficient roots through cell type-specific MTP8 expression for genetic complementation of *mtp8-1*. But several plant elements concentration under stress conditions can't be measured accurately by ICP-MS-FACS method. Elements, such as Ca and Mg, were enriched in cell wall under -K conditions. Thus, the discussion should include this limitation.

Response: We would like to thank the reviewer for the constructive comments.

We have now included one new dataset (new Suppl. Fig. 5) showing the detection limits of our method for each element with different numbers of protoplasts. If the procedure is carefully performed, paying attention to eliminate or significantly reduce sources of contamination, using adequate controls and a sufficient number of protoplasts, as we describe in the manuscript, accurate measurements are obtained. As pointed out by the reviewer, if the accumulation of certain elements in response to particular stress conditions or genetic perturbations is mainly confined to the apoplast, our method may not detect reliably their gradients. We now clarify this point in the Discussion (lines 367-368): "As our method determines the ionome of protoplasts, it detects concentration changes primarily associated with intracellular pools."

They found a strong accumulation of manganese in trichoblasts of iron-deficient roots (Fig. 4). Roots from 5-day-old seedling were analyzed in this study. Root hair has great contribution to ion uptake at this stage. To further support the conclusion, I think a root hair less mutant should be used as a control. It should be determined whether root hairs loss affects accumulation of manganese in trichoblasts.

Response: We thank the reviewer for the suggestion. Performing the experiment suggested by the reviewer would require the introgression of the *pEXP7::YFP* reporter in a root hair-less mutant, selecting homozygous mutants expressing the reporter and propagating a large number of these plants to produce enough seeds to perform FACS and ICP-MS. This would require at least 14 months. As an alternative approach to address the reviewer's suggestion, we inhibited root hair elongation chemically with L-kynurenine (L-kyn). As shown in new Suppl. Fig. 9a-f, we selected a concentration of

the inhibitor that was just sufficient to inhibit the extension of root hairs without negatively affecting *pEXP7* expression, trichoblasts specification, the longitudinal length of *pEXP7::YFP*-expressing epidermal cells or overall root growth. The size of *pEXP7::YFP*-expressing protoplasts isolated from roots of plants grown with L-kyn were significantly smaller than those isolated from mock-treated plants (Suppl. Fig. 9e). Importantly, qPCR analysis of Fe deficiency-induced genes, including *MTP8*, demonstrated that L-kyn did not significantly alter the expression of these genes (Suppl. Fig. 9f). We then grew *pEXP7::YFP*-expressing plants in Fe-limiting conditions with or without supplementation of L-Kyn to perform FACS and ICP-MS. The new results (Fig. 4h) show that the content of Mn in trichoblast cells is significantly decreased when root hair elongation is inhibited. We then investigated whether the smaller trichoblast size affected Mn partitioning between roots and shoots. As shown in new Fig. 4i-j, the efficiency of *MTP8*-mediated Mn accumulation in roots was significantly decreased when root hair extension was inhibited. The results are shown in Fig. 4 h-j and Suppl. Fig. 9, described in lines 312-327 and discussed in lines 397-402.

Minor comment:

Fig.1 b-f check the spelling error “Concentration”

Response: Thanks, done.

Reviewer #2 (Remarks to the Author):

The manuscript by Giehl et al. described a cell type-specific mapping of element distribution in *Arabidopsis thaliana* roots by the method that combines fluorescence-activated cell sorting and ICP-MS. It seems the accurate and precision determination of the mineral element in protoplasts of *A. thaliana* roots is one of the key steps. In this work, the analysis of the mineral elements was performed with two instruments: a sector field HR-ICP-MS and Q ICP-MS. The authors should provide the detailed information of the parameters of ICP-MS for the determination. The HR-ICP-MS with collision and reaction cell technologies is capable of analyzing most elements in the periodic table from lighter (Li), transition to heavy metals (U) simultaneously at levels of 10 pg/ml with a mass resolution of <1 amu because the matrix interferences are reduced or removed prior to detection. The related information such as isotopes, mass resolution, mass window, sample time, integration window, detection mode, etc. should be provided. However, when using Q ICP-MS, even with collision cell technique, the polyatomic interferences induced by the system and the complex matrix would be very severe especially in the mass range of 31–80, thus the precision determination of ^{56}Fe , ^{57}Fe , ^{64}Zn , ^{66}Zn , ^{67}Zn , ^{76}Se , ^{77}Se , ^{78}Se , ^{80}Se ect. are very difficult especially at very low concentration.

Response: We would like to thank the reviewer for the constructive comments.

Following the reviewer’s request, in the new Suppl. Table 3 we now provide a detailed description of the parameters used for HR-ICP-MS. The advantages of HR-ICP-MS listed by the reviewer were exactly the reasons why we used this system in all analyses, except for the experiment shown in Fig. 3b, which was analyzed with a Q ICP-MS. However, please note that in this specific dataset we only compared relative changes in the concentration of 9 elements in protoplasts isolated from wild-type and *hma4* mutant plants. Therefore, as the overall sample matrix was comparable, we expect that the contribution of possible interferences not sorted out by this system were comparable between the two sample sets.

Another difficulty for the cell-specific element analysis is the contaminant-free and integrity isolation of protoplasts from roots. Thus the images showing the integrity of protoplasts should be provided.

Response: Images of GFP-expressing protoplasts taken with a fluorescence microscope immediately after isolation and during simulated sorting were presented in Suppl. Fig. 4c. To further address the reviewer's comment, new Suppl. Fig. 1d now presents brightfield images showing the integrity of protoplasts isolated with the standard isolation method or the modified solution established in the present study.

Fig. 1 h,i were used to estimate the potential element leakage during isolation, "Compared to pure mannitol or the supernatant containing mock protoplasts, Li levels were approx. 2.3 times higher in the solution containing Li-enriched protoplasts", but figure 1h shows the opposite results, furthermore, the result indicated that Li was released from protoplasts.

Response: Thank you for picking up this mistake. The color code was incorrectly assigned. This mistake has now been corrected in the legend of Fig. 1h. In Fig. 1g, we now included a schematic to better illustrate what was analyzed. Higher concentrations of Li were indeed detected in the supernatant containing protoplasts isolated from Li-enriched roots. However, it is currently impossible to determine whether the Li detected in the supernatant of Li-enriched protoplasts was due to export from the protoplasts and/or originated from protoplasts that burst after isolation. We mention this in lines 193-194. To further estimate how much the elemental concentration in isolated protoplasts changes over time, we also determined the Li content in 100,000 protoplasts isolated from Li-fed plants. The new results (Fig. 1j) show that the Li content was comparable within 15 or 30 min after isolation, suggesting that no significant change occurred during this period. These results are described in lines 195-198: "We then quantified Li in 100,000 protoplasts sorted within different time intervals after isolation. Although Li contents slightly decreased when sorting was prolonged to 30 min, the decrease was not significant (Fig. 1j). Therefore, we restricted sorting to 30 min after protoplast isolation."

Thus, the obtained data show that all most all the detected elements including the Mg and P were relatively low enrichment in protoplasts compared with in the roots (Supplementary Figure 3), I doubted that it because of the leakage of mineral elements from protoplasts during extraction.

Response: As we mention in the Discussion (lines 347-351), the calculated elemental concentrations in protoplasts reported in the present study are in good agreement with results obtained using other methods such as LA-ICP-MS and quantitative x-ray (Chen et al., 2019 DOI: 10.1186/s13007-020-00566-9; Ryan et al., 2019 DOI: 10.1111/pce.13531). It is noteworthy that the HR-ICP-MS results from whole roots reflect elemental concentrations in protoplasts and apoplast (incl. cell walls and intercellular spaces) combined. The apoplast represents an important cellular pool of accumulation for many elements. For some elements, such as Ca, B and many of the transition metals, the apoplast represent even a main cellular pool of accumulation, thus it is expected that the concentrations of these elements are lower in protoplasts. In the case of Mg and P, more pronounced differences in concentration between protoplasts and whole roots were detected when plants were grown in K-depleted medium. For Mg, this appears to be especially associated to an increase of Mg in the apoplast, as Mg concentrations in protoplasts did not differ significantly between -K and control (Suppl. Fig. 3). We mention this in lines 150-152 and 159-161. In the case of P, there was a more dramatic decrease in protoplasts than in whole roots. Although we did not investigate this observation in details, it might indicate that P uptake is decreased under -K conditions as a way to

maintain the charge balance of root cells, which is affected by positively charged K^+ and negatively charged phosphate ions.

In addition, the unit of element concentration is $\mu\text{g/g DW}$, what is the meaning of “DW” ?

Response: “DW” refers to “dry weight” and “FW” to “fresh weight” of the tissue that was used for elemental analysis. These abbreviations have now been explained in the legends of the corresponding figures. Please note that, in plant nutrition, element concentrations in plant tissues are referred to unit of mass. Element content, in turn, is defined as the amount of an element per organ, tissue or, in our case, a specific number of protoplasts. Since it was difficult to accurately determine the dry weight of pelleted protoplasts, the results shown in Figs. 1b-f and Suppl. Figs. 3 and 4d are expressed as concentrations on fresh weight basis.

Collectively, the methodology used in the study is not innovative enough that could not provide high spatial compartmentation of the distribution patterns of important mineral elements in the plants.

Response: We cannot completely follow the reviewer’s comment. We agree that our method has some limitations, which we openly discuss in lines 366 and 385. However, we demonstrate that our method identifies significant differences in ion concentrations in different cell types (Fig. 2d), which can be tracked back to cell type-specific compartmentation mechanisms (Fig. 4). Thus, we believe the proposed method complements other existing cell type-specific omics-approaches and provides an important tool to map the concentration of multiple elements in specific cell types, especially when distinguishing between apoplastic and symplastic pools is critical.

Reviewer #3 (Remarks to the Author):

The authors describe the development of a cell-type specific approach to map the ionome in Arabidopsis roots. Optimizing FACS-ICP-MS for this purpose enabled them to generate a multi-element map of ion distribution throughout the layers of root tissue. They uncover a gradient of elemental distribution across the radius of the root. Furthermore, they find alterations of elemental distribution patterns upon disruption of xylem loading through two individual mutant lines. Finally, they found that trichoblast-specific loading of manganese is critical for the stress response to iron deficiency. Overall, this work provides an excellent example of how novel integration of traditional chemical and biological methods can lead to new discoveries and mechanistic understanding. In addition, it has important applications for understanding how plants acquire, store, and transport nutrients and toxins.

The data shown largely supports the conclusions of the manuscript. Overall, the methodology is sound and the methods are sufficiently detailed. Below are the minor suggestions I have for further improving the manuscript:

Response: We thank the reviewer for the encouraging and constructive comments.

1) Abstract, line 42: The authors describe how trichoblast Mn sequestration “efficiently retained manganese in roots and prevents toxicity in shoots.” The way it is written, it sounds as if the roots need the Mn. My understanding is that this retention is necessary to prevent Mn toxicity in the rest of the plant. If this is correct, I suggest rewording to: “Restricting manganese sequestration in

trichoblasts but not in endodermal cells efficiently retained manganese in roots, therefore preventing toxicity in shoots. These results hint to cell type-specific...”

Response: Thank you for pointing this out. Of course, plants need Mn as it is an essential element but if provided in excess, holding back Mn in roots helps to prevent Mn toxicity in shoots. We have rephrased the sentences as suggested by the reviewer but needed to replace ‘restricting’ by ‘confining’ to avoid that ‘restricting’ is misunderstood in the sense of ‘limiting’. Thus, the revised sentences now read (lines 41-45):” Confining manganese sequestration in trichoblasts but not in endodermal cells efficiently retained manganese in roots, therefore preventing toxicity in shoots. These results indicate the existence of cell type-specific constraints for efficient metal sequestration in roots. Thus, our approach opens a new avenue to investigate element compartmentation and transport pathways in plants.”

2) The optimized FACS-ICP-MS procedure relies on washing/incubating cells in 5% mannitol and or 5 mM salt, which are compounds frequently used to induce abiotic stress responses in plants (generally at higher concentrations, but protoplasts may experience stress at lower concentrations than intact plants). Do the authors see signs of stress in the cells after FACS? This could be assessed by measuring gene-expression levels in the ABA pathway. There are known links between ABA signaling and iron homeostasis, so if the optimized procedure is causing stress that may influence element levels in the cell, that should be discussed briefly in the manuscript.

Response: We did not attempt to maintain protoplast integrity after sorting, as collected protoplasts were not introduced into the ICP-MS system as single cells but first digested and then injected. We now clarify this point in the Material and Methods (lines 490-491). In our method, the protoplasts only come in contact with 5 mM NaCl during sorting, as this is used as the FACS sheath solution. However, osmotic stress induced by the much longer incubation in mannitol may be expected. We therefore estimated whether element concentrations in cells are changed during our procedure by analyzing Li concentrations in protoplasts isolated from Li-enriched roots. As shown in Fig. 1h-j, we did not detect significant changes in Li concentration up to 30 min of incubation in 5% mannitol. We present these new results in lines 195-198. We did not investigate further signs of osmotic stress responses in protoplasts. In principle, they could also involve Ca or ROS signaling, which may then not be specific for mannitol but a consequence of protoplastization.

3) Page 5, line 163: The logic of this paragraph is difficult to follow. 1) It is a little bit confusing to follow the *irt1* mutant experiments vs. the Fe deficiency experiments. I think this section and the discussion would be easier to follow if the term Fe-depleted media was used instead of “Fe deficient plants.” 2) I also don’t understand – if *IRT1* has poor selectivity for Fe²⁺, then is it more selective for Mn and Zn? If so, why doesn’t Mn decrease in the *irt1* mutant in +Fe conditions?

Response: We are sorry that our description was not sufficiently clear. To explain this experiment more accurately, we now replaced the term “Fe-deficient plants” by “Fe-depleted media”, as suggested by the reviewer. Regarding point 2), although *IRT1* can transport Mn and Zn, it is not the main uptake transporter for these micronutrients. The *IRT1*-dependent loading of non-Fe metals is only seen under low-Fe conditions, when *IRT1* is up-regulated and the protein takes up also other divalent metals if they are present in the growth medium. Under +Fe, Mn uptake is mainly facilitated by *NRAMP1* (Cailliatte et al., 2010 DOI: 10.1105/tpc.109.073023). Thus, the differential effects of Fe

supply and IRT1 function on the uptake of Fe, Mn and Zn provided an excellent basis to test how reliably our method can detect the expected changes in the concentrations of these micronutrients in protoplasts. We now describe these aspects more clearly in the text (lines 171-174): “When Fe was supplied to the media, only Fe and Zn concentrations were significantly lower in protoplasts isolated from roots of *irt1-1* than in protoplasts of wild-type roots (Fig. 1d–f), whereas a drop in Mn accumulation under this condition was probably prevented by the activity of the Mn uptake transporter NRAMP1^{35, 36}.”

4) What is your limit of detection with this method (in terms of cell number)? This information would fit into the FACS optimization sections. Right now, it is referenced in the discussion (Line 348-349) but the number is not present in the results section.

Response: Interesting point. To experimentally address the reviewer’s question, we performed HR-ICP-MS analysis of 11 elements on different numbers of FACS-sorted protoplasts. The limit of detection depends on the sample and the element that is analyzed. Our new experiment shows that at least 50,000 protoplasts are required for the reliable detection of up to 11 elements of interest. The new results are presented in the new Suppl. Fig. 5 and mentioned in the text (lines 211-212).

5) Line 201: Recommend rewording sentence to say, “...sorted samples if a contaminant-free cell sorter is not available...”

Response: Thank you. We amended the sentence as suggested.

6) It seems like characterizing a stele reporter line would provide further evidence about the enrichment of ions towards the inner root.

Response: Yes, we agree with the reviewer. However, in a previous attempt using the stele reporter *pWOL::GFP* and the pericycle reporter line E3754, we found that the estimated proportion of protoplasts expressing these cell markers was less than 4-6% of all sorted events. Consequently, we were not yet able to obtain sufficient protoplasts for reliable HR-ICP-MS. Since our current setup requires that at least 300 µL of solution is injected into our HR-ICP-MS instrument, the samples collected from the cell sorter (ranging from 30 to 150 µL) must be filled up to 300 µL with acidified water. Consequently, the less protoplasts sorted, the more diluted the sample will be. This is one of the reasons why we cannot yet reliably detect many elements in smaller numbers of protoplast. As we discuss in lines 379-385, we expect to be able to analyze more cell types once we have a better sample introduction system, allowing less or even no dilution of sorted samples.

7) In the section starting on line 239, the concept of xylem loading is briefly introduced. I think the authors should include a more thorough description of xylem loading, especially in the context of ion transport. I would not expect a biologist unfamiliar with plant biology to understand this section without more information about xylem loading.

Response: Thank you for the suggestion. To more clearly introduce the concept of xylem loading to a wider audience, we have now added the following sentence (lines 249-251) “In order to reach distant organs and tissues, mineral elements move radially across different cell layers in the root before they are loaded into xylem vessels via specialized ion transporters for long-distance translocation (Barberon and Geldner, 2014).”

8) Line 243: Should it read, “Heavy Metal ATPase 4 (HMA4)”?

Response: Yes. Thank you. We corrected it.

9) Line 255 – 258: The hypothesized relationship between citrate and Mn, Zn, and Ca should be explained.

Response: One possibility to explain the results shown in Suppl. Fig. 7e is that the constitutive upregulation of the Fe acquisition machinery in roots of the *frd3* mutant results in increased uptake of Mn and Zn, whose excess is moved on for long distance-transport. Since expression of the major constituents of the Fe acquisition machinery, i.e. AHA2, FRO2, IRT1 and the coumarin release, are confined (exclusively) to rhizodermal and cortical cells, metals accumulate in endodermal cells, in particular if their xylem loading capacity is not increased accordingly. Besides Fe, also Mn, Zn and to a lower extent even Ca can be transported in the xylem sap in the form of citrate complexes (Flis et al. 2016, doi.org/10.1111/nph.13964), which will depend ultimately on FRD3. We have added this piece of information into the text (270-271).

10) MTP8 should be italicized, this is done inconsistently throughout the manuscript.

Response: We carefully checked the manuscript and wrote MTP8 in italics when referring to the gene or transcript, and in regular font when referring to the protein.

11) Methods Section, line 380: Why did you use two different pSCR lines?

Response: This was because the two labs involved in the study started developing their methods independently. Only at a later stage we realized that the endodermal reporter introgressed into *hma4* was *pSCR::GFP* and not *pSCR::YFP*. Since *pSCR::GFP* is used only to compare elemental levels in endodermal cells of wild-type and *hma4* roots, we believe that the use of two reporters for this particular cell type does not affect the main outcome and conclusions of this study.

Reviewer #1 (Remarks to the Author):

Development a root hair-less pEXP7::YFP reporter mutant line costs time. The authors provided an alternative. Auxin biosynthesis inhibitor L-kynurenine was used to inhibit root hair. I think the revision has satisfactorily addressed my comments. Although the methodology used in the study is not innovative enough as the second reviewer said, I think their method provides an important way to investigate element transport pathways at cell level in plants.

Minor comment:

"Fig.2d pEXP7::YFP reporter", "Fig. 4e pEXP7::GFP", "Fig. 4H pEXP7::YFP". It makes me a little confused which reporter were used, or both?

The color mark of +Fe or -Fe should be wrong in the Supplementary Figure 9 (b, c, d, f), please check it.

Reviewer #2 (Remarks to the Author):

The work by Ricardo F. H. Giehl et al. has developed a novel approach that combined fluorescence-activated cell sorting (FACS) method and inductively coupled plasma mass spectrometry (ICP-MS) to investigate the immobilization profiles of mineral element in specific cell types of plant roots. The authors have provided new data in the revised manuscript to demonstrate the precision and detection limits of mineral element determination in different cell types. Moreover, the manuscript has been improved with the inclusion of experiment illustrations, making it easier to comprehend the purpose and significance of the results. Overall, the innovative approach of combining biological mutant sorting with precise analytical techniques is highly intriguing and has the potential to significantly advance our understanding of nutrient enrichment, heavy metal detoxification, and mineral element interactions in plants.

In perspective, the combination of FACS with single cell ICP-MS (scICP-MS) or high-resolution mass spectrometry imaging technology (such as laser ablation coupled ICP-MS) is promising and presents an opportunity to establish a transcriptomics-ionome network. I recommend that the authors make some comments on this in the manuscript.

Minor suggestion:

Please check the description in line161-163, "zinc (Zn) levels dropped in response to P deficiency (Supplementary Fig. 3b)", what I observed in the figure is the Zn levels elevated in "-P" groups.

Reviewer #3 (Remarks to the Author):

The authors have conducted a number of additional experiments to address previous comments, and relevant sections of the manuscript have been revised accordingly. Overall, the conclusions the authors make are supported by experimental evidence. In addition, this work is a valuable contribution to the development of a relatively accessible technology for measuring elemental composition within a tissue.

REVIEWERS' COMMENTS

Reviewer #1 (Remarks to the Author):

Development a root hair-less pEXP7::YFP reporter mutant line costs time. The authors provided an alternative. Auxin biosynthesis inhibitor L-kynurenine was used to inhibit root hair. I think the revision has satisfactorily addressed my comments. Although the methodology used in the study is not innovative enough as the second reviewer said, I think their method provides an important way to investigate element transport pathways at cell level in plants.

Response: We thank the reviewer for the encouraging comments.

Minor comment:

“Fig.2d pEXP7::YFP reporter”, “Fig. 4e pEXP7::GFP”, “Fig. 4H pEXP7::YFP”. It makes me a little confused which reporter were used, or both?

Response: Actually, we used both. The *pEXP7::YFP* line was used as the reporter line for trichoblasts. The *pEXP7::GFP* line was generated with the exact same promoter cloned to drive *MTP8* expression in *mtp8-1* complementing lines, and was used to verify the expected specific localization in trichoblasts. These points are now clarified in the Methods (lines 579-583) and in the legend of Fig. 4e (line 921).

The color mark of +Fe or –Fe should be wrong in the Supplementary Figure 9 (b, c, d, f), please check it.

Response: Thank you for spotting this mistake, which we now corrected.

Reviewer #2 (Remarks to the Author):

The work by Ricardo F. H. Giehl et al. has developed a novel approach that combined fluorescence-activated cell sorting (FACS) method and inductively coupled plasma mass spectrometry (ICP-MS) to investigate the immobilization profiles of mineral element in specific cell types of plant roots. The authors have provided new data in the revised manuscript to demonstrate the precision and detection limits of mineral element determination in different cell types. Moreover, the manuscript has been improved with the inclusion of experiment illustrations, making it easier to comprehend the purpose and significance of the results. Overall, the innovative approach of combining biological mutant sorting with precise analytical techniques is highly intriguing and has the potential to significantly advance our understanding of nutrient enrichment, heavy metal detoxification, and mineral element interactions in plants.

Response: We thank the reviewer for the encouraging comments.

In perspective, the combination of FACS with single cell ICP-MS (scICP-MS) or high-resolution mass spectrometry imaging technology (such as laser ablation coupled ICP-MS) is promising and presents an opportunity to establish a transcriptomics-ionome network. I recommend that the authors make some comments on this in the manuscript.

Response: We agree that this is an exciting perspective, which we already partially addressed in the Discussion. To highlight this aspect even further, we modified the sentence as follows (lines 403-

409): “We expect that this method will help to determine how individual transporters and their expression domains shape the ionome in different root tissues and cell layers, and finally determine nutrient partitioning to shoots. Furthermore, the possibility to combine our method with transcriptomics and to develop it further towards single cell ICP-MS paves the way to investigate transcriptome-ionome networks at very high spatial resolution.”

Minor suggestion:

Please check the description in line161-163, “zinc (Zn) levels dropped in response to P deficiency (Supplementary Fig. 3b)”, what I observed in the figure is the Zn levels elevated in “-P” groups.

Response: We are sorry for this misunderstanding. The reference to a decrease in Zn levels in P-deficient roots related to the data shown on Suppl. Fig. 3a, while the reviewer refers to the elevated relative Zn distribution shown in Suppl. Fig. 3b. Since the whole sentence reads (lines 161-163) “Interestingly, zinc (Zn) levels dropped in response to P deficiency and showed higher relative partitioning to the symplast” we should have referred to both panels of Suppl. Fig. 3. This is now corrected in the revised manuscript (lines 161-163).

Reviewer #3 (Remarks to the Author):

The authors have conducted a number of additional experiments to address previous comments, and relevant sections of the manuscript have been revised accordingly. Overall, the conclusions the authors make are supported by experimental evidence. In addition, this work is a valuable contribution to the development of a relatively accessible technology for measuring elemental composition within a tissue.

Response: We thank the reviewer for the encouraging comments.